# Care professionals' accounts of providing support and treatment for people with co-occurring alcohol use disorder and depression in the North East of England, UK: A qualitative study informed by complexity theory

Amy O'Donnell[1]*, Eileen Kaner[1], Barbara Hanratty[1], Eilish Gilvarry[1,2], Sarah Wigham[1,2], Katherine Jackson[1]

1 Population Health Sciences Institute, Newcastle University, Newcastle upon Tyne, United Kingdom,
2 Cumbria, Northumberland Tyne and Wear NHS Foundation Trust, St. Nicholas Hospital, Newcastle upon Tyne, Tyne and Wear, United Kingdom

* amy.odonnell@newcastle.ac.uk

## Abstract

### Introduction

There is an acknowledged care gap for patients with co-occurring substance use and mental ill-health. This study sought to use complexity theory to help make sense of the experiences of people who deliver or commission formal care for patients with alcohol use disorder and depression across one specific health and social care system.

### Method

Qualitative interviews with 26 health and social care professionals in the North East and North Cumbria Integrated Care System, England, were conducted. Data analysis was undertaken using reflexive thematic analysis and informed by key concepts from complexity theory.

### Results

Three main themes were identified: (1) how the interplay between risk, stigma and resource pressures influences how care professionals interpret and apply practice guidelines; (2) how individualised and disjointed practices have structural and historical roots, in particular the impact of health service commissioning cycles; (3) ways in which practitioners have been able to adapt and engage in creative practice to temporarily plug gaps in care.

**Data availability statement:** The data that support the findings of this study are available on request from Newcastle University Data Management Team at rdm@ncl.ac.uk. The data are not publicly available due to their containing information that could compromise the privacy of research participants.

**Funding:** This work was funded by a National Institute for Health and Care Research (NIHR) Advanced Fellowship (ADEPT: Alcohol use disorder and DEpression Prevention and Treatment, Grant: NIHR300616). The NIHR have not had any role in the design, implementation, analysis, write-up and/or dissemination of the research.

**Competing interests:** The authors have declared that no competing interests exist.

## Conclusions

The pressure of working with increasingly scarce resources, within a highly fragmented, shifting, and risk-averse care infrastructure, adversely affected professionals' capacity to provide consistent, patient-centered support. Innovations have emerged that address some of these barriers, but further investment is needed to better support the substance use and mental health workforce, including lived experience peer workers.

## Introduction

Alcohol consumption is causally related to over 60 different medical conditions [1], with drinking at harmful or dependent levels (alcohol use disorder, or AUD) [2] associated with adverse consequences that extend beyond people who drink excessively to their families, communities and society [3]. AUD is also commonly comorbid with a number of mental health conditions, including lifetime risk of major depression [4]; itself a leading cause of premature mortality and morbidity [5]. Up to 35% of harmful and dependent drinkers have a major depressive disorder or have experienced a major depressive episode over their lifetime, with lifetime prevalence of AUD in depressed individuals estimated at 40% [6].

Despite this evidence, there is widespread recognition that people with co-occurring depression and AUD frequently fail to receive appropriate care [7,8], with data showing markedly lower levels of treatment entry and utilization than those who have problematic substance use or mental ill-health alone [9]. Several factors contribute to this global care gap. Personal characteristics, including vulnerabilities related to the symptoms of heavy drinking and depression, can impair individual level motivation and capacity to initiate contact and/or engage with services [10,11]. People who use substances are found to have lower levels of health literacy, meaning that they may find it more difficult to communicate their needs for healthcare and understand healthcare information [12]. Cultural beliefs, the continued social and personal stigma attached to both mental ill-health and substance use, and lack of trust in the systems and providers of care, also present significant barriers to treatment-seeking [13]. Co-occurring AUD and depression is highly prevalent in populations facing multiple disadvantage [14,15], meaning that people experiencing these issues concurrently are also more likely to face material impediments (e.g., lack of stable housing, financial resources to pay for transport to appointments) to accessing support compared to those with single conditions, particularly in countries without universal health care [16,17].

Professional and structural barriers, arising from the political, legal, and administrative systems that frame health and social care provision, also affect the way care is delivered. Care providers are trained in professional siloes [18] and encounter variation in pre-qualification or registration requirements related to identifying, assessing and providing treatment for AUD. Post-qualification, training opportunities in alcohol and other substance use care may be available to non-specialist medical

and mental health care professionals, but these are often optional as opposed to compulsory [19,20]. General Practitioners (GPs) working in primary health care frequently cite lack of knowledge and appropriate training as a barrier to alcohol-related care [21], with evidence of limited delivery of brief interventions [22] and/or prescribing of pharmacotherapy for alcohol dependence in primary care [23]. Moreover, even specialist care professionals work with clinical protocols that are primarily designed to identify and treat single diagnoses as opposed to multiple conditions [24–27]. Implementation of these narrow and/or conflicting eligibility criteria can mean that some individuals are effectively "blocked" from accessing appropriate care; healthcare providers highlight that having an AUD frequently excludes patients from mental health support [11,28].

A further challenge concerns the complexity of the health and social care system that patients must navigate to access support for their mental health and substance use [29–31]. Existing evidence highlights how effective inter-organisational care provision can be undermined by incompatible information systems, poor leadership and communication, conflicting professional or organisational interests, and excessive bureaucracy [32,33]. In England, United Kingdom (UK), multiple providers are involved in the prevention, treatment of and recovery from co-morbid AUD and depression, with mental health and alcohol services currently commissioned (and regularly re-commissioned) by separate bodies (Integrated Care Boards versus Local Authorities respectively) [34]. The current care infrastructure spans services delivered by statutory (meaning care that the government is legally required to fund and provide) primary, secondary and tertiary level providers, as well as a range of non-statutory organisations (charities, community, and voluntary sector organisations). Navigating the resulting transitions between this constantly changing web of providers are risk-points for fragmented and/or incomplete care [11,35], and exacerbate the likelihood that patients will present with acute and/or uncontrolled symptoms in hospital emergency departments [36].

In previous work, we have explored the experiences of patients with co-occurring AUD and depression [31], these lived-experience perspectives are undoubtedly pivotal for understanding the current system. However, the perspectives and experiences of people who deliver or commission formal care for AUD and depression are also vital to inform the system improvements needed. A number of policy documents or commissioned reviews [8,37] have explored the perspectives of this latter group, most prominently in the UK, the recent Dame Carol Black report. Indeed, some of the challenges providers face might seem self-evident to those already working in this field of addiction and mental health. However, there remains a need to use appropriate theory when designing and conducting empirical healthcare research to ensure we produce more robust and generalizable evidence [38]. In this paper, our aim is to use complexity theory to help make sense of the experiences of people who deliver or commission formal care for AUD and depression, across one specific Complex Adaptive System (defined below) in the UK, with a focus on how these accounts might explain the challenges practitioners face in providing appropriate support. In the next section we introduce the theoretical approach before describing the methods.

## Theoretical approach

Substance use and mental health care provision, much like the wider health system, can best be understood by recognizing it as a Complex Adaptive System (CAS) [39]. A CAS comprises collections of individual agents, organisations, and sectors, who are nested within wider social, political and economic contexts and sub-systems, and whose actions are both interconnected and often unpredictable or nonlinear [40–43]. Thus, the micro interactions that occur between a patient seeking help for their alcohol use and mental ill-health (or more frequently, other related health concerns indicative of these conditions), and a clinician with the requisite skills to provide appropriate care, take place in a wider system of meso- and macro-level interdependencies that serve to constrain or enable particular care behaviours or outcomes [44]. However, whilst complexity can result in confusion and frustration on the part of those accessing and working within substance use and mental health services, it can also facilitate adaptation and flexibility on the part of both groups of system agents (patients and care providers), who may draw on their diverse knowledge, experience, and values, to navigate care access and provision across its fuzzy boundaries [45].

Complexity theory has been defined as '*a perspective that conceptualizes relationships of components…within a system as the foundation from which the properties of a system emerge*' [46]. Complexity theory is positioned as a response to more reductionist approaches to researching health systems that focus on their constituent components in isolation. In contrast, it strives to probe and characterize the complex behaviours that result from the interactions between those system components over time [47]. As such, complexity theory can help us to address questions that explore patterns of interaction between different components of a CAS, including individual care professionals ('agents'), policies, interventions, and their surroundings, which may occur at multiple system levels and times [48,49]. Through exploring the interactions, relationships, and feedback loops that exist between a diverse set of agents operating within a specific system ('self-organisation'), we can come to understand how they contribute to system change over time ('emergence') [30]. Complexity theory has been applied to various fields and methodological disciplines, including exploratory qualitative health systems research [46,50]. For example, Oakden et al used a complexity theory-informed lens to explore the challenges and opportunities of contracting out for public health and social services in Aotearoa New Zealand [51]; Cane et al explored women's help-seeking for problematic alcohol use drawing on complexity theory to inform data analysis [52].

## Methods

This qualitative study was conducted in the formative stage of ADEPT, a mixed-methods research project, which aims to develop an intervention to help improve formal care provision for people with co-occurring heavy alcohol use and depression [53]. We used semi-structured interviews to explore the views and experiences of people who provide services for co-occurring AUD and depression across one regional case study area. Our methods for data collection and analysis are described in detail below and we have completed a Consolidated criteria for reporting qualitative research (COREQ) checklist which is available in supplementary information (S1 File) [54].

### Study context

The case study context of the study is the North East and North Cumbria Integrated Care System. In the UK National Health Service (NHS) Integrated Care Systems are emerging infrastructures and an initiative of the NHS Long Term Plan that were introduced to support aligning provision of health and care across different organisations [55,56]. Integrated Care Systems comprise partnerships across NHS providers and commissioners, local authorities, community and voluntary services and their aim is the strategic planning, commissioning, and managing of care and facilitation of cross-organisational collaboration and integration across services that have historically been siloed [55,57]. Challenges to service integration within Integrated Care Systems have however been highlighted including incompatible information technology systems across different organisations and differing service cultures, e.g., between health and social care [58,59].

The North East and North Cumbria Integrated Care System serves a population of three million residents, making it one of the largest in England. Specifically it comprises a complex 'system of systems' across the following regions: North (Northumberland, North Tyneside, Newcastle and Gateshead), Central (County Durham, Sunderland and South Tyneside), Tees Valley (Hartlepool, Redcar and Cleveland, Middlesborough, Stockton on Tees and Darlington) and North Cumbria [60](ARC; NENC). Statutory health care across the North East and North Cumbria Integrated Care System is provided by 11 NHS Trusts, two mental health trusts, and 64 primary care networks, while public health and social care is provided by 13 local authorities. There are 13 adult drug and treatment services in the region including NHS specialist, third sector and primary care services [61]. Additionally, a wide range of non-statutory organisations (i.e., not part of the government funded care infrastructure) provide support for people with co-occurring AUD and mental ill-health. These include private (for profit) alcohol treatment providers, and Voluntary, Community and Social Enterprise organisations, some of which target individuals in general (e.g., Alcoholics Anonymous and SMART (Self Management and Recovery Training) groups), whilst others focus on specific populations such as families and carers (e.g., Adfam (Families, drugs and alcohol)), or people experiencing homelessness (e.g., Changing Lives).

## Sampling and recruitment

Our sampling frame was individuals involved in the delivery, design, and/or commissioning of services for patients with alcohol use disorder and/or depression across our case study system. Specifically, we aimed: 1) to include people who had a particular interest or knowledge about care for co-occurring alcohol use and depression to generate rich data on the topic; and 2) to ensure we had a geographic spread of participants and variation in terms of role, sector, condition and geographical or administrative area, as we were aware these factors might change their experiences of delivering care. We adopted a convenience sampling approach supplemented by snowball sampling, aiming to recruit approximately 25–30 individuals (or system agents). Recruitment started on 16th June 2021 and ended on 2nd February 2022. In the initial stages of recruitment, we identified potential participants through existing contacts, e.g., the North East and North Cumbria Integrated Care System Alcohol and Mental Health Working Group. Some potential participants (n = 8) were previously known to the interviewer while most were not. As the study progressed, we incorporated the snowballing approach, whereby participants sometimes suggested people to contact to invite to participate. For all interviews, we first contacted potential participants via telephone or email and provided the study participant information sheet; for those who expressed an interest in participating, we explained the study in more detail, provided an opportunity to ask questions, and emphasized the voluntary nature of taking part. If the potential participant was willing, we then gained written informed consent from participants.

## Data collection

Discussions were structured around the key questions of interest through the use of an interview topic guide (see supplementary information, S2 File). However, emergent/unforeseen issues were explored as appropriate, to allow us to consider the diverse professional experiences of the participants and to reflect our developing knowledge of how the case study care system has operated over time. The interview schedule was piloted on the first participant. No changes needed to be made, and this participant was included in the final sample. All interviews were carried out on a one-to-one basis by a post-doctoral female researcher (KJ), either using an online video conferencing platform (n = 21), over the telephone (n = 4) or face-to-face at their workplace (n = 1) and lasted a mean average of 58 minutes. All interviews were audio-recorded, transcribed by a professional transcription company, and checked for errors and fully anonymized by the researcher.

## Data analysis

Interview data were analysed using inductive and deductive reflexive thematic analysis in two stages [62,63]. The first stage of analysis initially ran in parallel with data collection until data collection stopped when theoretical data sufficiency was reached [64]. In line with Braun and Clarke's first three phases: 1) one researcher (KJ) familiarized herself with the data by reading and re-reading the interview transcripts, and making notes to capture immediate analytic observations and insights; 2), this researcher made notes on potential codes on each transcript while another researcher (AOD) made notes on potential codes on three of the transcripts (just over 10%), and they subsequently worked together to agree a coding framework The transcripts were then moved into NVivo Qualitative Research software (v21) to support data management, and the coding framework was applied to the data. 3) From here the two researchers worked together to generate initial themes to signify broader patterns of meaning and shared experiences across the dataset [63] Next, and analogous with the final three phases of reflexive thematic analysis, the first author (AOD) sought to: 4) further probe and develop these initial themes, focusing on identifying areas of resonance with concepts and principles from complexity theory, as outlined in Table 1 below, with an illustration in supplementary information (S3 File) [65].

The authors then further 5) refined and defined these themes to provide a coherent analytical story, which we present in the current paper, and which are further contextualized in relation to the wider existing literature (6). A central tenet of

**Table 1. Key complexity theory concepts that informed data analysis.**

| Concept | Description |
|---|---|
| Self-organisation | The way in which agents interact to coordinate their own forms, or patterns of behaviour arising from repeated agent interactions over time. |
| Feedback loop | Recursive mechanisms as a result of multiple agent interactions over time that create reciprocal behaviour either reinforcing (positive feedback loop) or undermining (negative feedback loop) each other. |
| Emergence | New system behaviours (larger entities) generated by the interactions of smaller or simpler entities. |
| Path Dependence | Past experiences influence the responses to new events. |

*Source: Adapted from Greer et al, 2018*

reflexive thematic analysis is recognition of how the researcher's positionality, subjectivity and skills inform the process of analysis and final interpretation. The researchers who undertook the interviews (KJ) and analysis (AOD and KJ) are female, White British, aged 40–49 and 50–59 years respectively, middle class and social scientists. Both researchers have extensive experience of conducting academic studies exploring care for people struggling with mental ill-health and/ or problematic substance use, but no direct personal experience of these issues. AOD has a specific interest in how theories, models, and frameworks from implementation and complexity science can inform our understanding of and response to these issues; KJ is particularly concerned with the health inequalities experienced by this population; thus, their interpretation and choice of theoretical lens was attentive to these foci. Input from a group of experts with relevant lived experience and established academic co-authors helped to further enhance the validity of the analysis.

## Ethics

The study was approved by an NHS Ethics Committee (North East – Tyne & Wear South Research Ethics Committee, reference 21/NE/0064).

## Results

Twenty-six professionals with experience of commissioning or delivering health or social care for people with AUD and depression were interviewed in total, representing statutory (government-funded) and non-statutory care providers (charities, community and voluntary sector organisations), as well as local government officials in commissioning roles in local authorities and the NHS (see Table 2). Three participants also noted unprompted during the interviews that they had personal experience of previous dependent alcohol or drug use.

Data analysis generated three main thematic areas related to practitioners' views of delivering care for people who drink heavily and have depression. The first two themes focus on how the negative feedback loops that exist between certain features of the existing health and care system, and the complex needs of this population, constrain professional practice. The third theme considers ways in which practitioners have been able to adapt and engage in creative practice to temporarily plug gaps in care over time through self-organisation. Illustrative quotes are presented below where possible and appropriate; we identify participants by job sector (i.e., primary healthcare) only to preserve anonymity. For one quote where a participant might be identifiable by job sector, we have removed any identifier.

### Risk, stigma, and capacity shape how the rules of care are applied

Many participants recognised substance use and mental ill-health as commonly co-occurring, which were also closely associated with people having experienced social-disadvantage and adverse life events. However, despite this shared understanding, interview narratives highlighted a fundamental dissonance between the complex social and clinical needs of service users, and

**Table 2. Participant characteristics.**

| Characteristic | | N |
|---|---|---|
| **Gender** | Man | 13 |
| | Woman | 13 |
| **Job Sector** | NHS Alcohol (inc. dual diagnoses within mental health teams and addictions psychiatry within alcohol services) | 10 |
| | NHS Mental Health (inc. recovery colleges and women's services) | 6 |
| | Primary healthcare | 4 |
| | Community and voluntary sector | 3 |
| | Local government | 3 |
| **North East and North Cumbria Integrated Care System geographical area** | Newcastle and/ or Gateshead | 6 |
| | Durham and/ or Darlington | 4 |
| | North Cumbria | 4 |
| | South Tyneside and/ or Sunderland | 3 |
| | Teesside | 3 |
| | Northumberland | 1 |
| | North Tyneside | 1 |
| | NENC region-wide | 4 |
| **Total =** | | **26** |

the rigid and singular nature of the care system infrastructure they had to work with. For example, there was a perception that current clinical practice guidelines obliged care professionals to separate out conditions for the purposes of treatment. This was particularly evident in the accounts of those working within statutory mental health services, where patients were often required to reduce their alcohol consumption before they could access support for their depression. As the following participant who worked in an IAPT service (Improving Access to Psychological Therapies, now known as NHS Talking Therapies) described:

> We operate off the IAPT national guidance with regards to substance misuse. So, we will work with people who are either using alcohol or substances, but I suppose it's that, sort of, level and impact, so what we class as probably moderate use. So, if somebody was, say, drinking heavily at a weekend, then it may well be that they're still suitable for our service, but if they were drinking, say, heavily every day of the week, then we'd probably be looking... to get their alcohol use managed more effectively first, prior to accessing therapy within our service. (P1 – NHS Mental Health)

Many participants discussed the exclusionary nature of these guidelines, with some noting that given certain individuals were unlikely to ever be fully abstinent, this effectively meant *'the doors are shut'* (P2) to mental health support. Whilst a few care professionals described being able to apply these guidelines more flexibly at times, their capacity to adapt their practice was increasingly limited by the financial pressures in substance use and mental health services:

> If the service is really, really pressured for capacity it is probably going to be easy to say, "We don't see someone that is intoxicated." If the service isn't as pressured at a point in time the chances are that they will slip through that barrier. I think there are a lot of variables that will impact on how bendy that line is. (P11 – NHS Mental Health)

Concern about risk and patient safety also appeared to influence how eligibility rules were applied across the system. Participants were acutely aware that self-harm and suicidality were highly prevalent within this patient group, but not all care professionals felt equipped to cope with mental health crises:

*... Somebody who is drinking really heavily and is suffering with depression, [might] struggle to engage consistently with us but we have some contact... then they, all of a sudden, come back in crisis and say, "I was drinking over the weekend and I was suicidal, I rang the crisis team, the crisis team wouldn't assess me because I'm under the influence, they wouldn't do anything." …The coordinator, at that point, gets scared like, "I don't feel safe enough to be working with somebody who was thinking about [suicide] at the weekend." Mental health services won't engage them, so then there's that fear of, "What do I do?" (P14 - Local Authority)*

As the above account suggests, such concerns related to both the safety of the patient and the professional, with other participants highlighting that some people who drink heavily and have mental ill-health could also be physically and/or verbally abusive. Whilst this represented a minority of patients, such negative experiences appeared to affect how people experiencing co-occurring AUD and depression were treated in general. Care professionals were further disincentivized to engaging with this population due to fear of organisational repercussions and the associated bureaucracy should they lose a patient to suicide. As Participant 14 went on to describe:

*It's such a shame that we do come from a fear base because unfortunately when we lose clients the first thing that happens is, "Right, we're going to quarantine the notes,.... we're going to do a death review, did you cover your risks, did you have a conversation with them about safe reductions?" (P14 – Local Authority)*

Some participants suggested that concerns around risk were exploited to justify passing on responsibility for the care of complex patients to others and reflected the continued stigma attached to substance use and mental ill-health. Whilst participants felt this was less evident in substance use focused services, incidents of substandard care were recalled from elsewhere in the care system. For example, one participant who worked in a substance use role in an acute hospital, described a recent such experience concerning a person experiencing homelessness who was drinking heavily, and the actions of other staff members:

*No one had done anything, they just literally wrote, "Left without treatment." and that was it... Yes, I know he was aggressive and could be difficult to manage but he still presented at ED obviously wanting help. (P15 – NHS Alcohol)*

Overall, despite the fact that UK health policy highlights the need to provide accessible and integrated care for this population, interview accounts conveyed how recursive – and predominantly negative – feedback loops between conflicting and risk-averse sector-specific guidelines, diminishing finances, and stigmatizing attitudes towards substance use and mental health, resulted in care pathways that were increasingly impenetrable and fragmented.

### Shifting and fragmented care delivery context shifts the burden of treatment

In addition to the challenge of working within the unsupportive and often conflicting procedural infrastructure outlined above, professionals also described the difficulties they encountered in providing joined up care in a care system where services, roles, and personnel altered frequently. This constrained their capacity to refer patients to appropriate providers and to develop the professional relationships or networks across the system that were viewed as critical to good care. As one participant explained:

*…what is not facilitating it is the constant change of drug and alcohol services providers. New provider every two years. Whenever you build up new relationships, they breakdown if commissioners decide to get a new service in place. (P9 – NHS Mental Health)*

Combined with a perceived lack of systemized training provision on AUD and/or mental health for care professionals, short term commissioning and organisational restructuring appeared to further limit their ability to retain relevant skills and knowledge. As well as destabilizing professional relationships within and across the care system, the resulting high staff turnover, particularly in secondary care and community alcohol services, meant expertise was often lost. Whilst people with co-occurring conditions were perceived as historically underserved, recent structural developments were viewed as having further reduced the capacity of the system to fully support their needs. Several participants with long term experience of working in this area of practice, highlighted the loss of dual-diagnosis roles as a specific example:

> There used to be specialist dual diagnosis roles. And the specialist dual diagnosis roles were, there was proper... dual diagnosis training. So, there was a skillset behind it. There was a knowledge base behind it. I think there was also something about an attitude behind it. The people that moved into these roles really understood and were in a position to address some of the stigma that is associated with problem drinking and challenge it. (P11 – NHS Mental Health)

A few participants working in primary health care suggested that a lack of appropriate specialist services for co-occurring conditions meant that patients attempting to access care from mental health services or secondary healthcare settings were often sent back to their GP for help. However primary care professionals' accounts suggested that they had limited capacity to support these patients, for example:

> If someone goes to Casualty with an overdose, and it's not that uncommon, we get a letter from the Psychiatric Liaison Team or from the Crisis Team, but that's pretty much putting it back to us, where it's co-occurring, saying, 'They were intoxicated. It was impulsive. Their lifetime risk of suicide is greater than the average population… They've often signposted them to alcohol services and also asked them to make an appointment with their GP. That's usually how that goes. (P5 – Primary Care)

Siloed and disconnected patient information systems could also undermine care professionals' attempts to work together in a coordinated way. In some areas, joined-up commissioning of services, for example where the community substance use provider had been commissioned by the relevant mental health trust, was perceived as having positively impacted care provision for patients with co-occurring conditions. However, these positive examples were uncommon, and vulnerable to being destabilised or ditched when services were re-commissioned. The more typical picture described by participants was of inadequate information sharing across and within organisations, with interactions with statutory services highlighted as particularly tricky potential 'blockers' to the flow of patient data:

> Information-sharing doesn't work brilliantly within the NHS, and it certainly doesn't work brilliantly between the NHS and social care, or between the NHS and the voluntary sector. That's a big impediment. (P3 – Voluntary, Community, and Social Enterprise)

In response to these gaps and inconsistences in the care infrastructure, participants described the emergence of care pathways in which individual patients were increasingly expected to take responsibility for their treatment to link up the chain of support. A range of formal and informal practices were highlighted as illustrating this shift in the burden of care. For example, care professionals conveyed that they relied on patients themselves to explain the reason for their referral to services. This meant that patients needed to be able to understand and communicate their care needs, often requiring them to repeat difficult experiences, as well as recall the range of services they were involved with. Additionally, participants cited the increased use of self-referral and signposting practices to connect patients to additional services as further evidence of this approach. Whilst for some participants these practices were viewed as helping evidence a patient's self-efficacy and readiness to change. However, several care professionals conveyed dissatisfaction with the use of

self-referral and sign-posting approaches with this patient population given the high risk of disengagement and the complex circumstances in which they often lived:

> *Community services are great, and they do a really great job, but I've found that if they [service user] haven't got an address or a phone, they won't accept a referral, which is infuriating because a lot of my patients are street homeless and don't have mobile phones. I've been letting some people come to the reception and taking the portable phone down so they can have that telephone assessment in the main entrance. If they don't attend appointments, if they don't attend twice, they get discharged, which I don't like, from community services. (P15 –NHS Alcohol Use)*

As the above account illustrates, these patients were particularly susceptible to missed-appointment penalties due to lack of engagement post-referral. Another participant based in the third sector discussed how they felt that current discharge practices were at odds with the non-linear recovery process:

> *…we would get individuals who would start to get well if their desire was to become sober or abstinent, like I say or just have some degree of stability in their drinking. They would start to engage with services, but, as soon as they went backwards in that journey, they naturally weren't turning up. They would just get letters sent out, and then they get discharged. Then they'd have to do the whole thing again. (P13 – Voluntary, Community, and Social Enterprise)*

## Plugging the gaps through relationships and adaptation

Despite the constraints to care described previously, some participants gave examples of how they were able to self-organise their practice to enable the delivery of more integrated and person-centered support. Evidence of such adaptive practices was most evident at the individual level where care professionals talked about how they drew on their relational capital and cross-system networks to overcome potential blocks in the care pathway. For example, one participant described how they would *'pick up the phone'* to their contact at the local drug and alcohol service to ensure they *'persevered'* in working with a more challenging patient (Participant 10, Secondary care – mental health). Another participant discussed how they exploited interpersonal connections (*'pulling in favours'*) to ensure they could access relevant patient information to support care provision (Participant 3 – Voluntary, Community, and Social Enterprise). Whilst most of these examples describe informal professional workarounds, one local authority had introduced a role with the explicit purpose of overcoming current system constraints. The care professional concerned described how they actively worked with other service providers to circumvent restrictive eligibility criteria that would otherwise prevent patients from accessing mental health support if they were drinking heavily:

> *My role, as I see it… is to break down those barriers and say, "Okay, the client may not be stable to engage with you right now, but your service is the one that they need. I can engage with you because the client's engaging with me. So, let us work together and you lay down your platform. You tell me what is expected. I will work with that client to meet those goals, and you can join them in the middle, and we'll come together. We'll wraparound the client." (P23 – Voluntary, Community, and Social Enterprise)*

Some other participants highlighted measures that senior management had introduced in their mental health service to address exclusionary practices for current heavy drinkers. For example, one care professional related the implementation of a policy that now required the crisis service to record information to explain why they have refused to see a patient who is drinking:

*We developed…a tool that they have to fill in even if they don't go out [to visit the patient], to write a clear rationale, what are the intermediate safeguards and risk management that they put in place?.... You have to at least put in some compassionate thinking into that case. Even if you decide not to go out, you have to still document your rationale, as opposed to bouncing it back beforehand or without documenting it." (P9 – NHS Mental Health)*

Increased recognition and integration of peer-to-peer support and relevant lived experience within the formal care system was highlighted by several participants as a further positive development. In this respect, participants drew comparisons between the limited more clinical skillset of care professionals working in statutory health services, and those based in Voluntary, Community, and Third Sector organisations, who were viewed as having more relevant 'soft' relational skills, such as empathy and compassion, often as result of their own personal experience of substance use or mental ill-health. All three of the participants who disclosed they had personal experience of alcohol or drug dependence reflected on how they had benefited from peer support and its role in helping others in a similar situation:

*The magic that happens when people get well doesn't come from the professional relationship. It is around a community of individuals who've experienced similar things, who are helping them. (Purposefully anonymized).*

Finally, whilst most shifts and adaptions in the system had emerged relatively gradually over time, the rapid onset of the COVID-19 pandemic was highlighted as having accelerated change in substance use and mental health service provision. Care professionals focussed on the rapid switch to digital health practices, both to facilitate inter-organisational working, as well as direct care delivery, as an example of this. However, although increased use of digital technology offered opportunities to increase parity of provision across the region and reduce pressures on practitioners working across a large geographical area with large caseloads, it had the potential for exclusion, thereby exacerbating inequalities, as the following participant outlines:

*Some people managed to use [iPads provided] appropriately and some people didn't use them appropriately. But I think for a certain number of people it was important. It gave us the capacity to keep in touch with them. I think overall our sense is it was worthwhile, even if there… are a few people where we just can't really get regularly in contact with them. Even when we offer them phones it doesn't work. (P8 – NHS Alcohol Use)*

## Discussion

### Main findings

Recognizing the rich insights that qualitative research can provide to our understanding of care provision in a complex health system [66], we used semi-structured interviews to explore the views and experiences of people who deliver care for patients with co-occurring AUD and depression across the North East and North Cumbria Integrated Care System [67]. In doing so, we drew on key concepts from complexity theory to probe the interactions and feedback loops that exist between individual care professionals, their employing organisations, and the wider socio-political contexts in which they work, to unpack how these shaped care provision. To our knowledge, this is the first application of this specific theoretical lens to understanding professionals' perspectives on the challenge of caring for this population.

We found that the interplay between several micro-, meso-, and macro-level influencers, created negative feedback loops that undermined practitioners' ability to deliver appropriate and/or seamless care for this patient population. At the individual service level, concerns around the consequences of failing to manage risk in an area of diagnostic complexity, which remains highly socially stigmatized and under-resourced, shaped care professionals' engagement with patients experiencing AUD and depression and led to inconsistency in how guidelines and eligibility criteria were understood and

applied. Yet even where patients were able to access support from one initial service, the odds that subsequent transitions along the care pathway would proceed smoothly appeared low. Despite the move to Integrated C are Systems in England [56], care professionals highlighted a lack of joined-up working within and across service providers. Alongside the physical, financial, and administrative separation of mental health and substance use services, the challenge of delivering care in a context of sustained structural change, with persistent and unpredictable commissioning cycles, was viewed as adversely affecting patient outcomes and experiences across most participants' accounts.

Previous studies have also highlighted the negative impact of health system fragmentation [29], budgetary constraints [68], lack of clear clinical guidance [24,25,28,69] and commissioning cycles [70] on care delivery for patients with co-occurring substance use and mental ill-health. In particular, there is a large existing literature linking the adverse consequences of sustained economic austerity in the UK on both the prevalence of mental ill-health and problematic substance use, and the capacity of services to support those struggling with these issues [8,71–74]. Recent evidence also demonstrates how siloed professional identities and organisational cultures continue to frustrate integrated care provision in England, with particular divergence noted between the clinical health and social care workforce [75]. Likewise, the existing literature highlights the potential for the increased focus on risk prevention across the NHS over recent decades to exert a pernicious influence on clinical practice [76–80], including for those working in statutory mental healthcare. For example, one study of mental health nurses identified how risk management processes led to several non-therapeutic defensive behaviours, including unnecessary onward referrals and a preoccupation with form-filling, and highlighted the daily struggle they experienced in balancing the needs of their patients against the need for self-protection [76]. Others have suggested that the rise in healthcare regulatory activity has ultimately shifted (or displaced) the initial primary target (managing risks to patients) to secondary concerns around avoiding political, legal and reputational risks to organisations or individual care professionals [79]. Additionally, research consistently finds that people with mental ill-health and/or problematic substance use may sometimes experience stigma and negative views from health care providers [81–83]. Further studies have highlighted the destructive nature of such relational stigma on patient help-seeking behaviors, and ultimately, health outcomes [31,84].

Here we use key concepts from complexity theory to advance our understanding of how the current care system for patients struggling with AUD and depression has been shaped by the interrelationships between these multi-level components over time. In doing so, we highlight how these recursive and negative feedback loops frequently result in this population receiving inadequate and inconsistent support and increasingly place the burden of care on vulnerable patients through a proliferation of individualized practices. Yet by scrutinizing participants' accounts for evidence of system adaption, we are also able to identify some more positive examples where new care practices have emerged in efforts to deliver more integrated, person-centered, and/or accessible support. Most of these examples represent informal adaptions or 'workarounds' adopted by individual care professionals, mostly through sheer persistence and/or by pulling in personal favours. However, we did identify some instances where more formalised approaches had been implemented in organisations, such as one local government organisation that has introduced a role with specific remit to tackle care barriers resulting from the application of unnecessarily rigid eligibility criteria.

Finally, whilst many of the adaptations identified in our study emerged over an extended period, the rapid onset of the COVID-19 pandemic represented an unprecedented 'shock' to the health and social care system in the case study region along with most of the world, particularly in terms of accelerating the move to digital technology. As others have also reported, we identified mixed views on whether increased digitisation of healthcare since 2020 has been positive, with particular concerns about the potential for digital exclusion for vulnerable individuals [85–88]. Given that certain elements of digital health transformation appear here to stay in the UK, with widespread adoption of a 'digital-first' approach in primary health care in particular [89], we share others' concerns about the potential impact on opportunities for help-seeking for patients with sensitive or stigmatized conditions, who may also face financial and digital literacy barriers to access [90].

## Strengths and limitations

Whilst we endeavored to recruit a range of care professionals from across the case study system, with different concerns and priorities, as we sought to interview people with a specific interest or expertise in co-occurring AUD and mental health, the sample may be unrepresentative of those who have less experience or interest in these conditions. As this was a qualitative study, we must stress that no inferences can be drawn about the prevalence of phenomena observed beyond the sample of participants we interviewed [91]. Also, as already noted, this study took place in a single care system (albeit covering a large and diverse geographical area) and focused on a potentially sensitive topic, meaning that some professionals might have felt uncomfortable talking about their practice due to anonymity concerns. However, the rich data that was generated suggests otherwise, with participants generally very candid about their experiences, both positive and negative.

A key strength of the study was our use of complexity theory to probe and interpret our data [46], an approach which in comparison to other theories, explicitly acknowledges the 'hypercomplexity', unpredictable, and essentially social nature of healthcare provision [49]. At the same time, we acknowledge the conceptual and practical challenges this approach presented, including how (and whether) we were able to identify relevant system agents (our care professional participants), or define meaningful boundaries and relationships across the system [92]. Additionally, whilst our conceptualization of the North East and North Cumbria Integrated Care System as a 'CAS' informed the design and conduct of this study, specific concepts from complexity theory were applied predominately post hoc during data analysis. As this theoretical lens did not guide the development of data collection tools, this might have limited the quality and comprehensiveness of the data we collected. Future studies could explore the potential of using complexity-framed methods such as social network analysis or system dynamics [93].

More fundamentally, social theorists have critiqued the relevance of employing concepts and ideas derived from the natural sciences to social contexts [94]. In contrast, we argue there is value in drawing on complexity theory to help us re-examine established areas of system 'failure'; it helps us gain more nuanced insights into what might otherwise be standard 'deficiency' explanations of poor or inconsistent care, whilst also encouraging us to uncover promising innovations and adaptations, as well as shared values, principles and practices, that can form the foundation of future improvement initiatives [44].

## Implications

Encouragingly, there is evidence of progress in some parts of the UK health and social care system that build on the promising practice identified in this study and previous calls for action [37]. For example, in response to Dame Carol Black's Review [8], a national commissioning quality standard (CQS) for local authorities in England is being implemented to support commissioning bodies to take a more strategic and coordinated approach to planning and provision of services, including for people with co-occurring conditions [95]. The UK Government have also launched the 'Addiction Mission', with the explicit aim of transforming the current drug and alcohol research ecosystem to accelerate development of innovative treatments and technologies [96]. There has been expansion in the implementation of more holistic treatment models to better support alcohol dependent patients to maintain recovery in the community, such as alcohol assertive outreach treatment and recovery navigators [97,98]. The rollout of Community Mental Health Transformation across England offers further opportunities to improve access to personalised and trauma informed care, including support for self-harm and coexisting substance use [56,99]. The updated guidelines for NHS Talking Therapies also now makes clear that drug and alcohol misuse are not automatic exclusion criteria [100]. In North East England specifically, the Integrated Care Board are developing a Joint Action Plan for Co-occurring Mental Health and Substance Use, with the ambition of making 'no wrong door' a reality for the region. Growing recognition of the valuable role that peers (people with relevant experiential expertise) can play in the recovery process also appears to be reshaping the health system, both directly, as part of

the physical care infrastructure, and indirectly, by influencing the skills and techniques that professionals based in formal services are using in their practice.

At the same time, successful implementation of these developments and innovations at system or service level still relies on having a professional workforce with the appropriate skills, resources, and motivation or agency to deliver them [55]. Currently, the generalist primary care workforce appears to be a key source of care for patients with co-occurring depression and AUD. Yet, primary care clinicians report feeling frustrated with few resources to help them [101]. Further, despite a 22% increase in the NHS mental health workforce between 2016–17 and 2021–22, there remain shortages of key clinical roles, with many services struggling to recruit and retain staff [102]. With evidence of a continued rise in mental ill-health and alcohol-specific mortality in England, particularly in the North East region [102–104], increased and sustained investment in substance use and mental health services is of course needed to address this shortfall. Although there has been a recent injection of additional funding in England [105], this will mostly help plug historic unmet need in the system rather than substantially increasing capacity [106]. Looking beyond the specialist treatment sector, efforts to better integrate support for this population across primary, secondary and tertiary care are also contingent on the capacity of regional health systems for strategic commissioning and meaningful cross-sector collaboration. There are concerns however that such ambitions are likely to be adversely affected by the recent cuts announce to Integrated Care Boards in the North East and North Cumbria and elsewhere [107].

The insights gained from the 'system agents' we interviewed suggest several areas for improvement to ensure we have a professional workforce that is better equipped to care for patients with multiple co-occurring needs. For example, the introduction of more flexible and relationally-based approaches to risk management [77], could help both foster trustful relationships with patients and better support clinicians to cope with challenging situations. Likewise, we would encourage freeing up care professionals to respond more creatively and compassionately to complex issues rather than being limited by rigid eligibility criteria, see for example the pioneering work conducted as part of Public Service Reform at Gateshead Council [108].

Finally, whilst clinical expertise remain critical to the mental health and substance use workforce with a clear need for increased investment in specialist treatment professionals [8], peer workers bring invaluable additional skills and lived experience to the care system. There is of course existing evidence of the benefits of using peer workers in mental health and substance use care, including positive impacts on treatment engagement including condition self-management, better relationships with formal care providers, and increased patient well-being and quality of life [109–111]. Yet, historically at least, peer workers have been low or unpaid care providers with limited opportunities for career advancement [112], and report experiencing stigma and discrimination from formally qualified professionals [113], as well as the ongoing challenge of maintaining the interpersonal boundaries critical to their own well-being [114]. Going forward, there is a need for improved recognition and support for these individuals [115].

## Conclusion

We used complexity theory to inform analysis and interpretation of qualitative interviews with 26 health and social care professional involved in commissioning and/or delivering care for people experiencing co-occurring AUD and depression in the North East of England, UK. We found that the interplay between several micro-, meso-, and macro-level influencers, led to multiple potential obstacles to delivering appropriate and/or seamless care for this patient population. The pressure of working with increasingly scarce resources, within a highly fragmented, shifting and risk-averse care infrastructure, adversely affected professionals' capacity to provide consistent, holistic, and patient-centered support. Innovative and adaptive practices have emerged across the health and social care system to address some of these barriers, but further investment is needed boost the substance use and mental health workforce and better support lived experience peer workers.

## Supporting information

**S1 File. Completed COREQ Checklist.**
(PDF)

**S2 File. Professional Interview Topic Guide.**
(DOCX)

**S3 File. Thematic Framework.**
(DOCX)

## Acknowledgments

Thanks to our Study Patient and Public Involvement Group for their ongoing and invaluable input into this research. We would also like to express our gratitude to the 26 care professionals who took part in the study and to those who helped with study recruitment across the NENC region.

## Author contributions

**Conceptualization:** Amy O'Donnell, Eileen Kaner.

**Data curation:** Amy O'Donnell.

**Formal analysis:** Amy O'Donnell.

**Funding acquisition:** Amy O'Donnell, Eileen Kaner.

**Investigation:** Amy O'Donnell, Katherine Jackson.

**Methodology:** Amy O'Donnell, Katherine Jackson.

**Project administration:** Sarah Wigham, Katherine Jackson.

**Resources:** Katherine Jackson.

**Supervision:** Amy O'Donnell.

**Writing – original draft:** Amy O'Donnell, Katherine Jackson.

**Writing – review & editing:** Amy O'Donnell, Eileen Kaner, Barbara Hanratty, Eilish Gilvarry, Sarah Wigham, Katherine Jackson.

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
