## [Decision Letter · Decision Letter 0]

19 Jul 2025

PONE-D-25-28708Care professionals’ accounts of providing support and treatment for people with co-occurring alcohol use disorder and depression in the North East of England, UK: a qualitative study informed by complexity theoryPLOS ONE

Dear Dr. O'Donnell,

Thank you for submitting your manuscript to PLOS ONE. After careful consideration, we feel that it has merit but does not fully meet PLOS ONE’s publication criteria as it currently stands. Therefore, we invite you to submit a revised version of the manuscript that addresses the points raised during the review process.

This is well-written manuscript that addresses an important and timely topic: care professionals accounts of providing support and treatment for people who have co-occurring alcohol use disorder and depression in one area in England. The value of the study is clearly articulated. The reviewers have raised important points to address in your revision, particularly around the role of complexity theory in the study (i.e. defining complexity theory and being clear how it informed your data collection and analysis) and an enhanced statement of author reflexivity. These are important considerations which I'd like to see addressed in the revised version. Some of the language from complexity theory was used throughout the results section (e.g. interconnections, adaptations), but  it wasn't always entirely clear to me how complexity theory influenced the data analysis and results presentation. For example, the Discussion section contains a discussion of feedback loops, but these weren't evident within the Results. Similarly, the limitations has a brief mention of defining meaningful boundaries, but there is no previous discussion about the process of making boundary judgements. I hope these comments and those from the reviewers are helpful as you revise and I look forward to receiving a revised and strengthened manuscript.

We look forward to receiving your revised manuscript.

Kind regards,

Elizabeth McGill

Academic Editor

PLOS ONE

Journal Requirements:

2. In the online submission form, you indicated that the data that support the findings of this study are available on request from the corresponding author (AOD). The data are not publicly available due to their containing information that could compromise the privacy of research participants.

4. Please remove all personal information, ensure that the data shared are in accordance with participant consent, and re-upload a fully anonymized data set.

Additional guidance on preparing raw data for publication can be found in our Data Policy (https://journals.plos.org/plosone/s/data-availability#loc-human-research-participant-data-and-other-sensitive-data) and in the following article: http://www.bmj.com/content/340/bmj.c181.long .

Additional Editor Comments:

Please see above.

Reviewers' comments:

Reviewer's Responses to Questions

**Comments to the Author**

1. Is the manuscript technically sound, and do the data support the conclusions?

Reviewer #1: No

Reviewer #2: Yes

2. Has the statistical analysis been performed appropriately and rigorously? 

Reviewer #1: N/A

Reviewer #2: Yes

3. Have the authors made all data underlying the findings in their manuscript fully available?

Reviewer #1: Yes

Reviewer #2: No

4. Is the manuscript presented in an intelligible fashion and written in standard English?

Reviewer #1: Yes

Reviewer #2: Yes

5. Review Comments to the Author

Reviewer #1: Dear authors,

Thank you for the opportunity to learn about your work.

The issue you address is of the utmost importance and studies like yours are definitely welcomed. But in its current form, this manuscript has two major flaws that needs to be fixed before we can assess its potential contribution.

In this manuscript, you aimed at offering key insights on how health care and social care services are capable of dealing with the complex case of patients struggling with both alcohol use disorders and depression. The originality of your contribution being that you adopted a complexity theory perspective. This is the key feature of your paper that had me accepting to review it.

Unfortunately, there is too little that is said on the complexity theory for your readers to know what your work is about. Having some good knowledge of the CT, I could have done without a description of it but the problem does not stop there. A major flaw in your manuscript is that you provide no clues on how the CT has informed data collection and analysis. These are the two main problems to which I add other minor ones that I summarise in the next few lines.

I am under the impression that your abstract does not align well with what I found in the paper.

For the reporting of your results, I can see that you follow most of the guidelines one can find, for instance in COREQ, however, I feel you need to make more visible how your statements/ observations are rooted in your dataset. For instance, writing “some participants” or “several participants” instead of “X participants (P1, P2, …)” cast some doubts on the rigour put into the analyses. There are a number of statements/observations that lack any reference to the data that makes them legitimate.

The use of acronyms is a bit of a pain, especially for your readers based outside UK. They should be avoided as much as possible. In table 1, for instance, I had to go back to the text to get what was the NENC ICS standing for. I also struggled with the VCSE.

I would suggest that instead of writing “Participant X”, when quoting your interviews, you use a short form like PX. The long form should be for the type of professional interviewed.

On page 5, line 68, I do not understand what you mean by: “particularly in countries with insurance-based health systems (16, 17).”

Page 11, line 215: “1) one researcher (KJ) familiarized themselves”… themselves or herself?

Page 25, line 502: “there is a large extant literature”… check whether this is the right way to say it.

To sum up my review, it is by far the problem of the application of the complexity theory to this topic that is most problematic. To be honest, I feel cheated having seen in the title reference to this theory but with nothing in the text that really makes it up to this promise.

Hope this will prove of some use.

Kind regards

A reviewer

Reviewer #2: Review comments

This is an interesting and well described study. The study provides important insights into service provision for people with co-occurring AUD and depression. The use of complexity theory to understand the results is a key strength of the research.

Line 154. Is there a reference for the larger background study?

Line 179: For consistency, can you provide a number for the ‘wide range’ of services that provide support for AUD and mental health?

Section 2.2: Please state the number of participants the study aimed to recruit

Section 2.3: Please state if individuals were individual or focus group interviews

Section 2.4: Please provide further detail in relation to how the coding framework was developed. This section states that all data were coded and then the coding framework was developed and applied to the data. If the data was already coded, why was the coding framework needed? What purpose did this serve?

Table 1 presents information on participant characteristics, the methods section states that recruitment was targeted to different professional roles. Including information on the role of participants is likely to be useful to the reader and help contextualise the findings - but, including this information needs to be balanced against the likelihood of possible jigsaw identification.

Discussion: Line 540: The section on peer support in recovery may be better placed in the implications section. Further discussion would be of benefit here to provide information on how peer services are being utilsed more broadly in the AUD / mental health treatment space.

Discussion: Consideration and discussion of how the results of the present study relate to the role of the ICS would be of use. Specifically, what implications do the results have for ICS service provision?

Lastly, the study uses reflexive thematic analysis, as such, a detailed reflexivity statement is needed. Some detail on the two researchers involved in data coding is presented in the methods section, but additional information is needed re how the researchers’ prior knowledge and experience may have influenced or shaped the research. This could either be situated in the method or discussion section.

6. PLOS authors have the option to publish the peer review history of their article (what does this mean? ). If published, this will include your full peer review and any attached files.

**Do you want your identity to be public for this peer review?** For information about this choice, including consent withdrawal, please see our Privacy Policy .

Reviewer #1: No

Reviewer #2: No

---

## [Author Response · Author response to Decision Letter 1]

29 Aug 2025

Please see detailed response to reviewers document attached.

---

## [Decision Letter · Decision Letter 1]

30 Sep 2025

Care professionals’ accounts of providing support and treatment for people with co-occurring alcohol use disorder and depression in the North East of England, UK: a qualitative study informed by complexity theory

PONE-D-25-28708R1

Dear Dr. O'Donnell,

We’re pleased to inform you that your manuscript has been judged scientifically suitable for publication and will be formally accepted for publication once it meets all outstanding technical requirements.

Kind regards,

Elizabeth McGill

Academic Editor

PLOS ONE

Additional Editor Comments (optional):

Dear authors

Many thanks for your revised manuscript. You have comprehensively addressed the reviewers' and my comments and the work has been strengthened as a result. I look forward to seeing this published.

Reviewers' comments:

Reviewer's Responses to Questions

**Comments to the Author**

1. If the authors have adequately addressed your comments raised in a previous round of review and you feel that this manuscript is now acceptable for publication, you may indicate that here to bypass the “Comments to the Author” section, enter your conflict of interest statement in the “Confidential to Editor” section, and submit your "Accept" recommendation.

Reviewer #2: All comments have been addressed

2. Is the manuscript technically sound, and do the data support the conclusions?

Reviewer #2: Yes

3. Has the statistical analysis been performed appropriately and rigorously? 

Reviewer #2: Yes

4. Have the authors made all data underlying the findings in their manuscript fully available?

Reviewer #2: No

5. Is the manuscript presented in an intelligible fashion and written in standard English?

Reviewer #2: Yes

6. Review Comments to the Author

Reviewer #2: Many thanks for your revised submission. All of the points in my initial review have been addressed. The revisions help to improve clarity and strengthen the paper. This is an interesting study that is well described and has clear implications for both policy and practice.

7. PLOS authors have the option to publish the peer review history of their article (what does this mean? ). If published, this will include your full peer review and any attached files.

**Do you want your identity to be public for this peer review?** For information about this choice, including consent withdrawal, please see our Privacy Policy .

Reviewer #2: No

---

## [Editor Report · Acceptance letter]

PONE-D-25-28708R1

PLOS ONE

Dear Dr. O'Donnell,

I'm pleased to inform you that your manuscript has been deemed suitable for publication in PLOS ONE. Congratulations! Your manuscript is now being handed over to our production team.

Kind regards,

on behalf of

Dr Elizabeth McGill

Academic Editor

PLOS ONE